# Generate More Imperceptible Adversarial Examples for Object Detection

Siyuan Liang [1]   Xingxing Wei [2]   Xiaochun Cao [1]

## Abstract

Object detection methods based on deep neural networks are vulnerable to adversarial examples. The existing attack methods have the following problems: 1) the training generator takes a long time and is difficult to extend to a large dataset; 2) the excessive destruction of the image features does not improve black-box attack effect(the generated adversarial examples have poor transferability) and brings about visible perturbations. In response to these problems, we proposed a more imperceptible attack(MI attack) with a stopping condition of feature destruction and a noise cancellation mechanism. Finally, the generator generates subtle adversarial perturbations, which can not only attack the object detection models that are based on proposal and regression but also boost the training speed by 4-6 times. Experiments show that the MI method has achieved the state-of-the-art attack performance in the large datasets PASCAL VOC.

## 1. Introduction

In recent years, a large number of research and applications have proven that deep networks can achieve state-of-the-art performance in various fields, including object detection(Lin et al., 2017)(He et al., 2017), semantic segmentation (Long et al., 2015)(Ronneberger et al., 2015), self-driving cars(Chen et al., 2015), face recognition (Shu et al., 2016)(Zhang et al., 2016), image-to-image translation (Isola et al., 2017)(Liu et al., 2017), etc. Although deep learning has achieved great success, recent studies (Szegedy et al., 2013)(Kurakin et al., 2016)(Madry et al., 2017) have confirmed that classifiers based on deep neural networks have serious security problems.

Object detection can be roughly divided into two categories: models based on proposal(Faster R-CNN (Ren et al., 2015)) and models based on regression(SSD (Liu et al., 2016)). As a critical subtask in computer vision, it is equally vulnerable to adversarial perturbation. The DAG (Xie et al., 2017) attacks the Faster R-CNN by optimizing to generate adversarial examples. The Bose's method (Bose & Aarabi, 2018) trains a generator to attack the face detector based on the Faster R-CNN. To attack proposal-based and regression-based models, the UEA (Wei et al., 2018) proposes a unified framework to generate a transferable adversarial example by combining the feature loss and the class loss.

However, these methods have apparent shortcomings. The DAG (Xie et al., 2017) serves as an iterative optimization method, which is very time-consuming. The Bose's method can not attack object detection models based on regression. The UEA method (Wei et al., 2018) has the following problems: Firstly, the generator needs to iterate at least fifty hours to make network convergence when the training images are 2511(VOC2007 (Everingham et al., 2007)). The time for the training significantly limits the extension of this approach to more massive datasets. Secondly, the unrestricted destruction of image features does not improve the black-box affect(the generated adversarial examples have poor transferability) and makes adversarial examples have excessive noise.

UEA method which is blindly minimizing the feature layer of the clean image and the Gaussian noise is unreasonable. We hope to find an adversarial example that is less noisy but still effective for two types of detection models. We call it the MI(more imperceptible) attack. The contributions of this paper can be summarized as follows:

• We have found that blindly destroying the target feature does not improve the transferability of adversarial examples.

• We find a stopping condition of the feature destruction. Under the same experimental conditions, we improve the transferability of adversarial examples and accelerate training speed by 4-6 times.

• For excessive noise, we use a noise cancellation mechanism(the group optimization and the random elimination) to generate more imperceptible adversarial perturbations.

[1]Institute of Information Engineering, Chinese Academy of Sciences, Beijing, China [2]Beijing Key Laboratory of Digital Media, School of Computer Science and Engineering, Beihang University, Beijing 100191, China. Correspondence to: Xiaochun Cao <caoxiaochun@iie.ac.cn>.

*Accepted by the ICML 2021 workshop on A Blessing in Disguise: The Prospects and Perils of Adversarial Machine Learning.* Copyright 2021 by the author(s).

## 2. Methodology

In this section, We will analyze the problems caused by over-destructive features in the UEA methods and introduce the MI attack approach.

### 2.1. Problem Definition

Given a image $x$, which have $n$ objects $O = \{o_1, o_2, ..., o_n\}$ in a specific dataset $D$. For objects in $ith$ image, it is represented by binary groups $\Gamma = \{(b_1^i, y_1^i), (b_2^i, y_2^i), ..., (b_n^i, y_n^i)\}$, where $b_n^i$ represents the coordinates of the $nth$ object in $ith$ image, and $y_n^i$ represents the label of the $nth$ object in $ith$ image. Assuming that object detection models are $M = \{m_1, m_2, ..., m_j\}$, then object detection can be expressed as $M : R^{w \times h \times c} \longrightarrow R^{|\Gamma|}$. The $M$ set includes both proposal-based models and regression-based models.

For an arbitrary detection model $m_j$, the correct classification label $f_y(x, o_n)$ and the bounding-box function $f_b(x, o_n)$ denote the detection results of the $nth$ object on the clean image $x$. We add a slight perturbation $\delta$ to generate an adversarial example $x + \delta$. When the adversarial example makes the labels of all objects go wrong or the IOU between predicted boxes and the ground truth is less than 0.5, $i.e.$, $\forall n$, $f_y(x+\delta, o_n) \neq f_y(x, o_n) \vee IOU(f_b(x+\delta, o_n), f_b(x, o_n)) < 0.5$, we think object detection model $m_j$ can be fooled successfully.

For adversarial attacks, we conduct the white-box attack on the Faster-RCNN model. In the training phase, we know the network architecture and parameters of the Faster-RCNN to train our generator. In contrast, we only know the output of the SSD model. Therefore, the adversarial attack for the SSD model is the black-box attack. The transferability of the adversarial example refers that the adversarial example has a good performance on the black-box attack.

### 2.2. Problems caused by feature destruction

In general, the transferability of the adversarial example for the black-box model has poor performance. For mainstream object detection methods, they utilize the deep neural network as a feature extractor. The Sabour's (Sabour et al., 2015) approach shows that the image features in a deep neural network (DNN) can be manipulated to generate adversarial examples. Thus, we can destroy image features to improve the transferability of adversarial examples for mainstream object detections, e.g., We can manipulate the image features to make it close to Gaussian noise.

**Problem 1** Do not improve the black-box affect(the adversarial example have poor transferability)

**Problem 2** Generated adversarial examples bring about visible perturbations.

**Problem 3** The training takes a lot of time.

$$\rho = \sum_{t=1}^{T+1} \frac{\theta}{T+1} \cdot \nabla J(x_t) - \sum_{t=1}^{T} \frac{\theta}{T} \cdot \nabla J(x_t) \quad (1)$$

To minimize the loss function, we need to find a trade-off between image similarity and feature destruction. The Eq. 1 represents the gain of the noise amplitude, and the $T$ represents the number of iterations. Eventually, image similarity will tend to be stable, which means that the disturbance will have a minimum upper bound $\theta$. The higher the iterations $T$, the smaller the gain $\rho$. In other words, the increment of each iteration $\rho$ is inversely proportional to the total number of iterations $T$.

### 2.3. Stopping Condition for the Feature Destruction

In the previous section, excessive feature destruction is not only useless for improving transferability and brings about evident perturbations. Thus, finding a suitable stopping condition is essential. Instead of iterating 20 times for each sample's feature loss, we use the SSD model as an indicator, and it will indicate to the stop of the feature destruction. We only need to send each generated adversarial example to the SSD model to get the labels of targets $y_{ssd}$ in the input. If $y_{ssd} \notin y_{ground-truth}$, then we think that the feature destruction is enough to attack the regression model.

To generate an adversarial example quickly, we still use a GAN to generate an adversarial example. The generation network is used to generate adversarial examples that can fool the classifier. The discriminator takes an image as input and attempts to predict it as the original image or the output image of the generator. It determines that the real picture is 1, and the generated picture is 0.

$$L_{GAN} = E_{x \sim P_{data}}[logD(x) + log(1 - D(x + \langle G(x), A \rangle))] \quad (2)$$

The GAN loss can be represented by Eq. 2, $D$ stands for discriminator, and $G$ stands for generator. By minimizing GAN loss, we can make the generated adversarial examples as close as possible to the spatial distribution of real images. Besides, for the detection task, we only need to perturb the critical target area so that the target area can not be detected. Therefore, during the training and testing process, we can get an attention matrix A through the Faster R-CNN.

$$L_{Sim} = E_{x \sim P_{data}}[\| \langle G(x), A \rangle \|_2] \quad (3)$$

We use the $L_2$ distance as a metric, by minimizing the $L_2$ distance between the adversarial example and the input

picture, as shown in Eq. 3.

$$L_{Cls} = E_{x \sim P_{data}} [\sum_{n=1}^{N} [f_{y_n}(x, o_n) \tag{4}$$
$$- f_{y'_n}((x + \langle G(x), A \rangle), o_n)]]$$

The Eq. 4 define the classification loss function. Where the $x$ represents the input and the $o_n$ can be defined as the $nth$ target area in the picture $x$. $y_n$ and $y'_n$ represent the classification label for the $nth$ target area on the clean image and the adversarial image. The $f$ represents the network of object detection.

$$L_{Fea} = E_{x \sim P_{data}} [\sum_{i=1}^{N} \| \langle A_i, (X_i - GN_i) \rangle \|_2] \tag{5}$$

In Eq. 5, the $X_i$ represents the $ith$ image features of network and the $GN_i$ represents the $ith$ Gaussian noise features. The $A_i$ represents attention matrix A in the $ith$ layer in deep neural network. The (Wei et al., 2018) achieves the purpose of destroying the feature of the image object region by minimizing the $L_2$ distance between the input and the Gaussian noise feature layer.

$$L_{total} = L_{GAN} + \alpha L_{Sim} + \beta L_{cls} + \gamma L_{Fea} \tag{6}$$

$$such \ that f_{ssd}(x + \langle G(x), A \rangle) \neq y_{ground-truth}$$

Therefore, the final loss function can be expressed by Eq. 6. It consists of GAN loss, similarity loss, classification loss, feature loss, and iteratively optimizes under the condition that the generated adversarial example can still effectively attack the SSD model. We set $\alpha$ is 0.05, $\beta$ is 1. We choose the fifteenth layer and the twenty layer in the feature loss, and the $\gamma$ is [0.00010, 0.00020]. The learning rates of the generator and the discriminator are 0.0002. We train the generator and the discriminator on a single GPU Titan XP. Under the same condition of the hardware, the adversarial examples have good attack performance on the mainstream object detection model, and the training time significantly shortens by 4-6 times compared with the UEA method.

### 2.4. Eliminate Noise Redundancy

When we use the Eq. 6, we can prevent over-destruction features to a certain extent and generate less-noise adversarial examples. However, the generated adversarial examples still have excessive noise. Similar to the method (Shi et al., 2019), we can get an adversarial example which has a smaller visual difference with the input by using the noise cancellation technique, and it can still attack the proposal-based model and the regression-based model. There is a better adversarial example $x_4$ around $x_3$, which not only

attacks both detection models at the same time but is closer to the original sample $x$.

$$max(\| x_3 - x \|_2 - \| x_4 - x \|_2) \tag{7}$$

The Eq. 7 represents the objective function of the optimal adversarial example $x_4$. Where $x$ represents the original image, $x_3$ represents the suboptimal adversarial example, and $x_4$ represents the optimal adversarial example. We only need to optimize on suboptimal solution $x_3$ in the previous section. Removing too much noise will cause the adversarial example to fail, so the optimal solution $x_4$ exists in a sphere with a radius of $\Psi$ in the center of the suboptimal solution $x_3$.

**Group Optimization**

$$z_0 = x_3 - x \tag{8}$$

The initial noise $z_0$ can be represented by Eq 8, and the number of iterations of the group optimization is $T_g$. We can divide the noise into $T_g$ groups and define the noise group $Z = z_n, z_{n-1}, ...z_1$, where $z_n = n/T_g * z_0, n = 1, 2, ...T_g$. In the test phase, we send clean images to the generator to generate the adversarial example $x_3$ and to the two types of detection models to get the initial labels $Y$. Next, we obtain the initial noise $z_0$ and calculate the noise group $Z$. The noise group $Z$ is sequentially added to the clean image and sent to the two types of detection models to obtain the adversarial classification labels $Y'$. When $\exists y \in Y', y \in Y$, stop iteration and return the last group noise $z_g$.

**Random Elimination**

$$z_r = z_g \cdot R, R = \begin{cases} 0, & random() \leq \pi \\ 1, & random() > \pi \end{cases} \tag{9}$$

The noise after group optimization is called $z_g$, but the noise still has some redundancy. Therefore, we adopt a random elimination method. Assuming that the random iteration optimization number is $T_r$, we initialize a matrix $R_0$ that is all 1 and randomly selects an element with a ratio of less than the threshold $\pi$ to be 0. The resulting new matrix $R$ is then multiplied by the group noise $z_g$ to obtain a new noise $z_r$, as shown by Eq.9. Next, the noise obtained from each iteration is added to the clean picture and sent to the object detector to obtain the adversarial classification labels Y'. When $\exists y \in Y', y \in Y$, the iteration stops and returns the previous noise.

## 3. Experiments

### 3.1. Datasets

For object detection in the image, we used 5011 images from the PASCAL VOC 2007 training set to train the generator. We will evaluate our approach from three perspectives: success rate, time, and image quality. **Success Rate:**For object

*Table 1.* Ablation experiment of MI method on images for Faster R-CNN model and SSD model. The ORI represents that only the SSD is used as a training stop condition to generate adversarial examples. +GO represents that adversarial examples are generated by using the group optimization on adversarial examples generated by the ORI. +RE represents that adversarial examples are generated by using the random elimination on adversarial examples generated by the ORI. MI represents that both the training stop condition and two noise cancellation methods are used to generate adversarial examples.

| Methods | UEA | ORI | +GO | +RE | MI |
|---|---|---|---|---|---|
| PSNR | 28.65 | 28.65 | 30.04 | 29.05 | **30.15** |
| mAP_FR | 0.21 | **0.11** | 0.14 | 0.14 | 0.16 |
| mAP_SSD | 0.06 | **0.02** | 0.05 | **0.02** | 0.06 |

*Table 2.* Comparison of UEA and MI on clean images.

| Evaluation | SSD | FR | Times(s) | PSNR |
|---|---|---|---|---|
| Clean Images | 0.70 | 0.68 | \ | Inf |
| UEA | 0.21 | **0.06** | 8.4 | 28.65 |
| MI | **0.16** | **0.06** | **1.8** | **30.15** |

detection, we use mAP to judge our attack effect. **Time:** The time here is the time to train the generator. **PSNR:** we evaluate the image quality by using the Peak Signal to Noise Ratio, PSNR.

### 3.2. Ablation Study

We discuss the ablation study of this method. +GO, +RE, MI respectively represent that only the grouping optimization algorithm is used on the basis of the original method, only use the random elimination algorithm, and use both of them. In Table 1, after using the noise cancellation algorithm, the image quality will be significantly improved, but the attack effect on the SSD will also decrease, which means that the image quality is negatively correlated with the transferability of adversarial examples. The grouping optimization algorithm is better than the random elimination algorithm. In the same attack effect, the group optimization algorithm generates adversarial examples which have better image quality. When we use both of them, we will get adversarial examples that have the best image quality, and its attack on SSD and FR will be worse.

### 3.3. Results on Adversarial Images

We mainly compare the performance of the UEA method and the MI method on clean images. As shown in Table 2, the adversarial examples generated by the MI method are very effective for the Faster R-CNN model and the SSD model. Compared with the UEA method, the mAP of adversarial examples generated by the MI method is only 0.16 (5 percentage points lower) for the black-box attack on the SSD model. Thus, the adversarial examples by the MI method have better transferability compared with the UEA

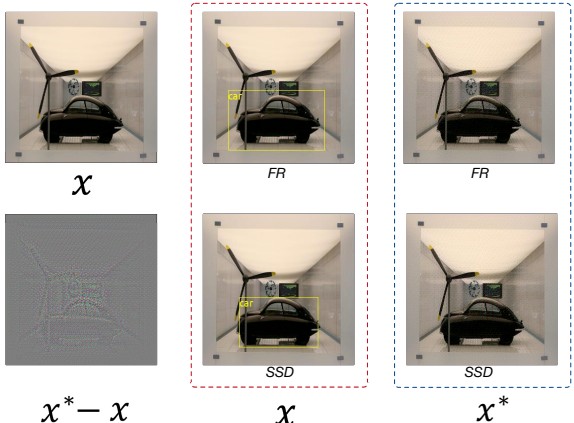

*Figure 1.* Quantitative results for images. The first line represents the original image, the detection results of Faster-RCNN in the original image, and the detection results of SSD in the original image. The second line represents the adversarial perturbations, the detection results of Faster-RCNN in the adversarial image, and the detection results of SSD in the adversarial image.

method. For the training time, the average time for the MI method to train a sample is only 1.8s, which is nearly 5 times faster than the UEA method. The shortening of training time makes it possible to extend the MI method to training on large datasets. For PSNR, the image quality generated by the MI method is higher than that of the UEA method. Therefore, the MI method generates more imperceptible adversarial examples, which are still valid for the Faster R-CNN model and the SSD model.

In Figure 1, the first and second columns show the results of the Faster R-CNN model and the SSD model on the clean image $x$ and the adversarial example $x^*$, respectively. $x^*-x$ represents the adversarial perturbations by the MI method. As shown in the Figure 1, the adversarial examples generated by the MI method can successfully fool both types of detection models.

## 4. Conclusion

In this paper, we propose an attack method that can generate more imperceptible perturbations to the clean images and effectively attack two types of object detection methods. Our proposed MI method shortens the training time of the UEA method by 4-6 times, making it possible to train the generator on a large dataset. At the same time, it can produce an adversarial example with higher image quality or more transferable. We have experimentally validated on large datasets PASCAL VOC and ImageNet VID. Considering the speed of training and the effect of the adversarial attack together, we believe that the MI method is superior to the existing adversarial attack methods for object detection on images.

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
