# OpenReview forum: "Generate More Imperceptible Adversarial Examples for Object Detection"
_ICML.cc/2021/Workshop/AML — ICML 2021 Workshop AML Poster_

### Official Review · Reviewer_c4mn · 2021-06-20
**Review for generative adversarial attack on object detection**

**Rating:** Accept
**Confidence:** 4

**Review:**

This paper develops a method to attack objective detection models by a generative model. The attack method uses a GAN to generate adversarial noise. The opimization process is done in the latent variable space. Authors also adapt early stopping method and two noise elimination methods.  The proposed method achieves higher transferability and invisibility compared to the baseline UEA method.

There are some quesitions unclear for me:

1. Compared to the baseline method, proposed method generates noise rather than the whole adversarial example. Apart from that,  loss function looks fairly similar to the baseline method. Is there any significant difference in loss function?  Whether the optimization object of generative models has big influence on transferability?
2. There should be a sketch of loss curve. Does early stopping improve the transferability by underfitting the current model?

In summary, this paper manages to improve and speed up attack against objective detection. It would make contributions to the workshop.

---

### Decision · Program_Chairs · 2021-06-21

**Decision:**

Accept (Poster)

**Comment:**

This paper developed a method to attack objective detection models by a generative model. It can be further improved by addressing the reviewer's comments.